# Activation of Epstein–Barr Virus’ Lytic Cycle in Nasopharyngeal Carcinoma Cells by NEO212, a Conjugate of Perillyl Alcohol and Temozolomide

**DOI:** 10.3390/cancers16050936

**Published:** 2024-02-26

**Authors:** Hannah Hartman-Houstman, Steve Swenson, Radu O. Minea, Uttam K. Sinha, Ming-Fu Chiang, Thomas C. Chen, Axel H. Schönthal

**Affiliations:** 1Department of Molecular Microbiology & Immunology, Keck School of Medicine, University of Southern California (USC), Los Angeles, CA 90089, USA; 2Department of Neurosurgery, Keck School of Medicine, USC, Los Angeles, CA 90089, USA; 3USC/Norris Comprehensive Cancer Center, Los Angeles, CA 90033, USA; 4Department of Otolaryngology, Keck School of Medicine, USC, Los Angeles, CA 90089, USA; 5Department of Neurosurgery, Fu Jen Catholic University Hospital, New Taipei City 24352, Taiwan; 6NeOnc Technologies, Inc., Los Angeles, CA 90069, USA

**Keywords:** endoplasmic reticulum stress, herpes virus, reactive oxygen species, chemotherapy

## Abstract

**Simple Summary:**

Nasopharyngeal carcinoma (NPC) is a cancer that is frequently associated with Epstein–Barr virus (EBV) infection, and this virus is thought to play a role in NPC development and malignant growth. Within tumor cells, the virus generally remains in a latent stage that supports cell survival and stimulates tumor growth. However, it has been postulated that it might be beneficial as part of cancer therapy to coax the virus out of its latent stage and into a lytic mode, which would result in tumor cell killing. A few agents are known to accomplish this switch, but their clinical success so far has been mixed. We are developing a novel compound, NEO212, that has shown anticancer activity in several preclinical tumor models. Here, in this report, we demonstrate that it is able to coax EBV into its lytic cycle, resulting in NPC cell death. We propose that NEO212 should be investigated further for its potential clinical usefulness as a therapeutic agent for NPC.

**Abstract:**

The Epstein–Barr virus (EBV) is accepted as a primary risk factor for certain nasopharyngeal carcinoma (NPC) subtypes, where the virus persists in a latent stage which is thought to contribute to tumorigenesis. Current treatments are sub-optimal, and recurrence occurs in many cases. An alternative therapeutic concept is aimed at triggering the lytic cycle of EBV selectively in tumor cells as a means to add clinical benefit. While compounds able to stimulate the lytic cascade have been identified, their clinical application so far has been limited. We are developing a novel anticancer molecule, NEO212, that was generated by covalent conjugation of the alkylating agent temozolomide (TMZ) to the naturally occurring monoterpene perillyl alcohol (POH). In the current study, we investigated its potential to trigger the lytic cycle of EBV in NPC cells in vitro and in vivo. We used the established C666.1 cell line and primary patient cells derived from the brain metastasis of a patient with NPC, both of which harbored latent EBV. Upon treatment with NEO212, there was an increase in EBV proteins Zta and Ea-D, key markers of the lytic cycle, along with increased levels of CCAAT/enhancer-binding protein homologous protein (CHOP), a marker of endoplasmic reticulum (ER) stress, followed by the activation of caspases. These effects could also be confirmed in tumor tissue from mice implanted with C666.1 cells. Towards a mechanistic understanding of these events, we used siRNA-mediated knockdown of CHOP and inclusion of anti-oxidant compounds. Both approaches blocked lytic cycle induction by NEO212. Therefore, we established a sequence of events, where NEO212 caused reactive oxygen species (ROS) production, which triggered ER stress and elevated the levels of CHOP, which was required to stimulate the lytic cascade of EBV. Inclusion of the antiviral agent ganciclovir synergistically enhanced the cytotoxic impact of NEO212, pointing to a potential combination treatment for EBV-positive cancers which should be explored further. Overall, our study establishes NEO212 as a novel agent able to stimulate EBV’s lytic cycle in NPC tumors, with implications for other virus-associated cancers.

## 1. Introduction

Nasopharyngeal carcinoma (NPC) is the most prominent cancer of the nasopharynx, arising from the mucosal epithelium to form a malignant neoplasm. Although rare in the United States, NPC is endemic to North Africa, the Middle East, Southeast Asia, the southern regions of China, and Taiwan. Three histopathological subtypes can occur: type I is comprised of keratinizing squamous cells, type II of non-keratinizing squamous cells, and type III of undifferentiated cells. Each subtype has unique genetic and epigenetic profiles, leading to varying levels of carcinogenicity and mortality. Subtypes II and III are associated with Epstein–Barr virus (EBV) infection [1,2]. 

EBV (also known as human gammaherpesvirus 4, HHV4) causes infectious mononucleosis and is associated with a number of non-malignant and premalignant diseases, as well as with malignancies such as Burkitt’s lymphoma, B cell lymphoma of immunocompromised patients, Hodgkin’s disease, occasionally gastric cancers and T cell lymphomas, and NPC. Upon the primary infection of cells, EBV follows one of two life cycles, namely, lytic (productive) infection or latency. While lytic infection results in the demise of the host cell, latent infection supports long-term persistence of the virus and, in some cases, oncogenic transformation of the cell [3,4]. 

In its latent form, EBV is present inside the cell as an episomal, circular DNA of approximately 172,000 base pairs encoding 85 genes. Very few genes are expressed at this stage. In epithelial cells, such as those developing into NPC, the main proteins are EBV nuclear antigen 1 (EBNA-1), latent membrane proteins (LMPs), and EBV-encoded, small non-coding RNAs (EBER). These gene products not only suppress the lytic cycle but are also involved in the tumorigenic process, assisting immune escape and stabilizing telomeres. Based on these pro-oncogenic mechanisms during latency, it has been contemplated that activation of the lytic cycle selectively in EBV-positive tumor cells could be part of a therapeutic approach to eliminate malignant cells. However, despite an increasing number of agents showing potential for such lytic induction therapy, only very few are being tested in clinical trials, with varying degrees of success [5,6].

Reactivation of the EBV lytic cycle is a complex process. At the gene expression level, it is defined by the sequential activation of immediate-early, early, and late genes. The earliest transcript that emerges is Zta (also called ZEBRA and EB1; encoded by the BZLF1 gene), along with another immediate-early protein, Rta (encoded by the BRLF1 gene). Both Zta and Rta are master transcription factors required for the initiation and completion of the lytic cascade. Among the subsequent early transcripts is Ea-D (encoded by the BMRF1 gene), which is transcriptionally activated by Zta and Rta. In total, about 80 EBV proteins participate in the lytic process, and Zta, Rta, and Ea-D represent convenient markers to document and investigate its early events [7,8,9]. 

A variety of different stimuli have been recognized to disrupt EBV latency and trigger its lytic cycle, although not all stimuli work equally well and might display preferences between EBV-infected epithelial and lymphoid cells. Among them are histone deacetylase (HDAC) inhibitors, DNA demethylating agents, calcium ionophores, phorbol esters, anti-immunoglobulin, transforming growth factor beta, and several chemotherapeutic agents. The selective reactivation of EBV in tumor cells could conceivably provide new targets for immunotherapy, make the cells more susceptible to antiviral drugs such as ganciclovir (GCV), or possibly result in tumor lysis [5,7,10]. 

GCV and some of its derivatives are used as antiviral medications primarily for the management of cytomegalovirus (CMV) infections. Like EBV, CMV belongs to the herpes virus family, and the two viruses display common structural characteristics [11,12,13]. Upon its uptake by cells, GCV is converted to 5′-monophosphate by a viral kinase, i.e., thymidine kinase of EBV or its homolog UL97 in CMV (collectively referred to as TK) [14,15]. This first phosphorylation step by TK is followed by two further phosphorylation events that are performed by cellular kinases. The resulting GCV triphosphate disrupts viral DNA replication by competing with 2′-deoxyguanosine triphosphate for incorporation into the growing DNA strand. Because GCV triphosphate lacks a 3′-OH group, DNA elongation cannot proceed once this nucleotide has been incorporated; additionally, GCV triphosphate preferentially inhibits viral DNA polymerases more than cellular DNA polymerases [13,16,17]. 

The above-described antiviral effects of GCV cannot take place during the latent stage of herpes virus infection, because the expression of viral TK remains suppressed. Consequently, the rationale for cytolytic virus activation (CLVA) therapy envisions that triggering the lytic cycle—which goes along with the stimulation of TK expression—would result in the sensitization of virus-infected cells to the inhibitory impact of GCV [14,15,18,19]. The validity of this principle has been demonstrated in a number of preclinical models, and there is preliminary evidence that it might exert clinical benefit as well (e.g., [14,18,19,20,21,22,23,24,25,26]). Overall, while these results are highly encouraging, CLVA therapy remains to be fully developed; in particular, it requires further discovery and characterization of suitable drug combinations to achieve optimal therapeutic outcomes for EBV-positive epithelial and lymphoid tumors. 

In our prior work, we designed a novel chemotherapeutic agent, called NEO212, that was created as a conjugate of two anticancer agents, temozolomide (TMZ; an alkylating agent) and perillyl alcohol (POH; a naturally occurring monoterpene) [27]. TMZ is, in clinical use, part of the standard of care for the chemoradiation treatment of glioblastoma (GBM). Its primary cytotoxic mode of action is methylation of the O6-position of guanine, which results in DNA double-strand breaks and subsequent apoptosis [28]. However, this process does not work in cells that express the DNA repair protein O6-methylguanine-DNA methyltransferase (MGMT), because MGMT is able to effectively remove the methyl group from the O6-position of guanine, thereby disposing of the toxic lesion and providing chemoresistance [29]. Not surprisingly, TMZ is little effective against MGMT-positive tumors [30,31]. 

POH is a metabolite of limonene and is present in the essential oils of citrus fruit peel, peppermint, sage, cherries, and other botanicals [32]. Its anticancer activity has been documented in a large number of preclinical studies, but several clinical trials with an oral formulation of POH were not successful, because gastrointestinal toxicity proved dose-limiting and therapeutic activity was unimpressive [33,34,35]. However, an intranasal formulation of POH, called NEO100, was found to be easily tolerated and has provided some preliminary indications of better activity in patients with GBM [36,37,38]. 

NEO212, the bioconjugate of TMZ and POH, has shown promising anticancer activity in a large number of preclinical tumor models, including NPC [39,40,41,42,43,44]. However, it has never been investigated in EBV-positive NPC cells, nor is it known whether NEO212 might be able to trigger the lytic cycle and sensitize tumor cells to GCV. We, therefore, investigated these issues and, in the following paper, present the results of these studies. 

## 2. Materials and Methods

### 2.1. Pharmacological Agents

NEO212 was synthesized by Axon MedChem (Reston, VA, USA) as an off-white crystalline powder and was kindly provided by NeOnc Technologies, Inc. (Los Angeles, CA, USA). It was dissolved in DMSO to a 100 mM stock solution. TMZ (Sigma Aldrich, St Louis, MO, USA) was dissolved in DMSO to a concentration of 50 mM. 06BG (Santa Cruz Biotechnology, Santa Cruz, CA, USA) was dissolved in DMSO to a concentration of 50 mM. Aliquots of these stock solutions were stored at −80 °C. POH was obtained as NEO100, a highly pure version of POH which was produced under current good manufacturing practice (CGMP) conditions by Norac Pharma (Azusa, CA, USA) and provided by NeOnc Technologies, Inc. It is an oily liquid that was stored at −4 °C. It was diluted fresh before each use with DMSO to a 600 mM stock solution. In all cases of cell treatment, the final DMSO concentration in the culture medium never exceeded 0.5% and was much lower in most cases. Hydrogen peroxide was obtained from a CVS Pharmacy as a 3% stabilized topical solution (USP). Ascorbic acid (Sigma-Aldrich) was prepared fresh before use by dissolving powder in phosphate-buffered saline (PBS) to a stock concentration of 1 M. Beta-mercaptoethanol (Sigma-Aldrich) was used fresh and diluted with culture medium. 

### 2.2. Cell Culture

Human NPC cell lines TW1 and TW4 were kindly provided by Prof. Chin-Tarng Lin (National Taiwan University Hospital, Taipei, Taiwan) [45]. Human NPC C666.1 cells were kindly provided by Prof. Kwok-Wai Lo (The Chinese University of Hong Kong, Hong Kong SAR, China) [46]. Human T98G and U251 glioblastoma cells and human Raji lymphoma cells were sourced from the American Tissue Culture Collection (ATCC). Except for the Raji cells, all cells were cultured in Dulbecco’s Modified Eagle Medium (DMEM; prepared with raw materials from Cellgro/MediaTech, Manassas, VA, USA), which was provided by the Cell Culture Core lab of the USC/Norris Comprehensive Cancer Center. DMEM was supplemented with 10% fetal bovine serum (FBS; Omega Scientific, Tarzana, CA, USA and X&Y Cell Culture, Kansas City, MO, USA), 100 U/mL penicillin, and 0.1 mg/mL streptomycin (both from Gibco/Thermo Fisher Scientific, Waltham, MA, USA). The Raji cells were cultured in RPMI instead of DMEM. The cells were kept in a humidified incubator at 37 °C and a 5% CO_2_ atmosphere.

### 2.3. Primary Cells (BM-NPC)

The use of patient-derived, de-identified, fresh tumor tissue was approved by the Institutional Review Board (IRB) of the University of Southern California. A freshly resected brain metastasis, provided by Dr. Thomas Chen, from a patient with NPC was rinsed with PBS, cut into small pieces with a scalpel, and digested with trypsin for 20 min at 37 °C. The cells were centrifuged, and the pellet was suspended in DMEM growth medium supplemented as described above. The cells were able to proliferate for a few passages. 

### 2.4. Immunoblots

Cells were harvested by centrifugation, and cell pellets were lysed with RIPA buffer (Thermo Fisher Scientific) containing protease and phosphatase inhibitors. After the determination of protein concentration, we used the Western blot procedure to analyze the lysates according to previously published protocols [47]. The primary antibodies were as follows: In order to detect the MGMT protein, a polyclonal antibody from Cell Signaling Technology (Danvers, MA, USA; #2739) was used. For cleaved caspase 3, we used a monoclonal antibody (MAB10753) from Millipore-Sigma (Burlington, MA, USA). For EBNA-1, Ea-D, Zta/ZEBRA, CHOP, TRA-1-81, and beta-actin, we used monoclonal antibodies (sc-81581, sc-58121, sc-53904, sc-166682, sc-21706, and sc-47778), which were from Santa Cruz Biotechnology. As secondary antibodies, we applied horseradish peroxidase-antibody conjugates (Jackson ImmunoResearch Laboratories, West Grove, PA, USA) according to the supplier’s recommendations. All immunoblots were repeated at least once to confirm the results. Uncropped images of Western blots along with molecular weight markers and the densitometric scanning of images are provided in the Appendix A.

### 2.5. Colony Formation Assays

Depending on the cell line and the plating efficiency, 600–800 cells were seeded into each well of a six-well plate and treated as described previously [39]. Drugs were added the next day. After 7 days, the medium was replaced with a drug-free medium. After 14–21 days, the newly formed colonies (groups of >50 cells) were stained with methylene blue (1% in methanol) and counted. CFAs were always performed in duplicate or triplicate and, in most cases, were independently repeated.

### 2.6. Transfection with siRNA

Cells were placed in six-well plates in serum-free medium and transfected using the Lipofectamine 2000 Kit (Invitrogen/Fisher Scientific, Waltham, MA, USA) according to the manufacturer’s instructions. Two different siRNAs were used in parallel: one targeting CHOP, the other targeting green fluorescent protein (GFP) as a control. Both were obtained from Ambion/Fisher Scientific) and used at five nanomoles for the knockdown. After transfection, cells were collected, and protein expression was analyzed by Western blot. 

### 2.7. Animal Model

The in vivo experiments presented in this manuscript were approved by the IACUC of the University of Southern California (Institutional Animal Care and Use Committee of USC) prior to the initiation of any work with animals. Balb/c athymic nude/nude mice (Harlan, Inc., Indianapolis, IN, USA) were implanted into the right flank with 10^7^ C666.1 cells. After the xenografted tumor cells had developed a palpable tumor, the mice were randomly separated into different treatment groups. The control group received the vehicle only (45% glycerol, 45% ethanol, 10% DMSO). NEO212 and TMZ were first dissolved in DMSO, then diluted in ethanol and glycerol to match the vehicle composition. The subcutaneous tumor volume was measured with calipers every 5 days; the body weight was monitored; and the overall survival was recorded. 

For the analysis of tumor tissue by Western blot, the cells were implanted as above. Once subcutaneous tumors reached a volume of about 1000 mm^3^, the mice were treated with 50 mg/kg NEO212 or a vehicle. One group of mice received a single dose and was euthanized 24 h later; another group of mice received a second dose 24 h later and was euthanized at 48 h. Tumor tissue was excised and flash-frozen. For the extraction of proteins, the frozen tissue was placed into a mortar, along with a small amount of liquid nitrogen to keep the tissue frozen and hard. The tissue was ground with a pestle until homogeneous, then transferred into a microcentrifuge tube with a stainless steel micro spatula. The RIPA buffer was added, and the lysate was processed for the determination of protein concentration and Western blot analysis. 

### 2.8. Calculation of Drug Combination Effects

Cells were treated with increasing concentrations of individual drugs (i.e., NEO212 or GCV), as well as their combinations, and the resulting cell toxicity was determined by CFA. Drug combination efficacy was evaluated with the Bliss independence model [48], which focuses on treatment effect enhancement. This method compares the observed combination response (Y_O_) with the predicted combination response (Y_P_). Typically, the combination effect is declared synergistic if Y_O_ < Y_P_. 

### 2.9. Other Statistical Analyses

All parametric data were analyzed using Student’s t-test to calculate the significance values. For animal survival, the log-rank (Mantel–Cox) test was used. A probability value (*p*) < 0.05 was considered statistically significant. 

## 3. Results

### 3.1. Drug Sensitivity of NPC Cells and Correlation with MGMT Levels

Three different human NPC cell lines—TW1, TW4 and C666.1—were analyzed by colony formation assays for their sensitivity to growth inhibition by NEO212 and TMZ. As presented in Figure 1, NEO212 prevented colony formation with an IC50 of 28, 41, and 11 µM in TW1, TW4, and C666.1 cells, respectively. In comparison, treatment with TMZ up to 100 µM had a negligible effect and did not reach IC50. The inclusion of the MGMT inhibitor O6BG potently sensitized all three cell lines to TMZ, indicating that MGMT was protecting the cells from the cytotoxic effects of this drug. The addition of O6BG to NEO212 showed a similar sensitizing effect in TW1 and TW4 cells. But this was not the case in C666.1 cells, presumably because these cells already were highly sensitive to NEO212 even in the absence of O6BG. 

Because the NEO212 molecule consists of POH conjugated to TMZ, we next investigated whether a mere mixture of POH plus TMZ would be able to mimic the potent cytotoxic effect of the fusion product. Cells were treated with NEO212, TMZ, POH, and POH mixed with TMZ at equimolar concentrations. As shown in Figure 2, the combination treatment with POH and TMZ as individual agents was unable to replicate the potent toxicity of NEO212. For instance, while 60 µM NEO212 killed close to 100% of TW1, TW4, and C666.1 cells, a combination treatment consisting of 60 µM POH plus 60 µM TMZ had no effect in any of the cell lines. These results indicate that NEO212 exerted a much stronger growth-inhibitory potency than the sum of its parts.

Based on the strong chemosensitizing effect of O6BG, we sought to confirm the presence of MGMT in these cells by Western blot analysis. Figure 3 shows that all three NPC cell lines were positive (densitometric scanning of Western blots is provided in the Appendix A). The C666.1 cells displayed the highest MGMT levels, which was intriguing because, in the above-shown CFAs (Figure 1), the inclusion of O6BG was unable to sensitize these cells to NEO212. Also included in this MGMT Western blot was a lysate made from primary cells derived from the brain metastasis of a patient with NPC; these cells (called BM-NPC) were successfully cultured for a few limited passages and showed positivity for MGMT. 

### 3.2. NEO212 Triggers Immediate-Early- and Early-Stage Lytic Viral Protein Production

In the above, the response of the C666.1 cells to treatment with NEO212 was unusual in that their IC50 was much lower than the IC50 of the TW1 and TW4 cells, and, despite the presence of copious amounts of MGMT, O6BG was unable to confer increased sensitivity. Because C666.1 cells are known to harbor EBV, we performed a Western blot analysis of the EBNA-1 protein to document a potential differential between the different NPC cell lines. As expected, the C666.1 cells tested strongly positive for EBNA-1, whereas the TW1 and TW4 cells were negative (Figure 3). The BM-NPC cells also showed clear positivity. As a positive control, we included Raji Burkitt’s lymphoma cell lysate. Raji cells are well recognized for being EBV positive, although their EBNA-1 protein is known to be a variant of a smaller molecular size [46]. 

Having confirmed the presence of EBV in some of the NPC cells, we next pursued the hypothesis that NEO212 might be able to trigger the lytic cycle of EBV. As indicators of the lytic cycle, we chose Zta and Ea-D proteins, which are EBV proteins not expressed during the latent stage which emerge during the initiation of the lytic stage. The cells were treated with NEO212 and analyzed by Western blot. Figure 4A,B show that NEO212 caused a prominent, persistent induction of both lytic cycle markers at concentrations as low as 1 µM, which was followed by increased levels of pro-apoptotic cleaved caspases 3 and 7. The induction of these lytic cycle proteins by NEO212 could also be documented in primary patient-derived BM-NPC cells (Figure 4C). In comparison, TMZ was unable to trigger the lytic cycle (Figure 4D). As a negative control, we used TW1 cells, and, as expected, no EBV proteins could be detected upon their treatment with NEO212 (Figure 4E). 

### 3.3. NEO212 Exerts Therapeutic Activity and Triggers the Lytic Cycle In Vivo

We next investigated some of these events in vivo. Mice were subcutaneously implanted with C666.1 cells and, once palpable tumors had formed, were treated with oral NEO212, TMZ, TMZ + POH, or a vehicle once daily for five consecutive days. Tumor development was continuously monitored, and, once the tumor volume had reached 1500 mm^3^, the animals were euthanized. The C666.1 cells turned out to form very slow-growing tumors. Yet, over the course of 150 days, all the animals not receiving NEO212 had to be euthanized due to large tumor masses (Figure 5). In comparison, none of the animals treated with NEO212 reached a tumor size of 1500 mm^3^. The difference in survival between the vehicle-treated control group and the NEO212-treated group was highly significant (*p* = 0.0006). In contrast, none of the other treatments (TMZ alone or combination treatment with TMZ mixed with POH) achieved a statistically significant difference to the vehicle-only treatment. Thus, NEO212 demonstrated anticancer activity that was superior to TMZ or TMZ mixed with POH. 

In a separate experiment, C666.1 tumor tissues were collected from mice one day and two days after oral treatment with NEO212 and analyzed by Western blot for the presence of EBV proteins. As displayed in Figure 6A, the tumors derived from animals that had been treated with NEO212 showed a clear induction of lytic marker Zta protein, along with the prominent emergence of cleaved caspase 3. At the same time, the EBNA-1 levels were strongly down-regulated. Taken together, these results established that NEO212 was able to induce the lytic cycle of EBV under in vivo conditions as well. 

### 3.4. CHOP Is Required for the NEO212-Induced Lytic Cycle

In an effort to determine the mechanism by which NEO212 induces EBV’s lytic cycle, we investigated a few potential candidate pathways and came across CHOP, a key protein of the endoplasmic reticulum (ER) stress response [49]. As shown in Figure 6A, the treatment of animals with NEO212 strikingly stimulated the expression of this protein. We confirmed this effect in vitro as well: the treatment of cultured C666.1 cells with NEO212 caused a clear induction of CHOP expression (Figure 6B). Notably, this increase came before the increase in lytic cycle proteins Zta and Ea-D, suggesting that CHOP might be involved in stimulating their expression. 

To illuminate a potential causative role of CHOP, we used siRNA to knock down and prevent its expression. This was followed by the treatment of cells with NEO212 and an analysis by Western blot. As shown in Figure 7, the suppression of CHOP expression resulted in the blockage of Zta expression, indicating that CHOP was required in order for NEO212 to trigger the lytic cycle. 

### 3.5. ROS Are Involved in the NEO212-Induced Lytic Cycle

A number of cellular stress conditions are known to trigger ER stress, among them increased levels of reactive oxygen species (ROS). We investigated the potential role of ROS by including anti-oxidant compounds to suppress ROS production. C666.1 cells were treated with NEO212 in the presence or absence of ascorbic acid (AA) and beta-mercaptoethanol (BME), two well-known ROS scavengers [50,51]. Figure 8A,B shows that NEO212 was neither able to potently stimulate CHOP expression nor trigger the lytic cycle when AA or BME were present in the cell culture medium, indicating the requirement of ROS for these processes. 

A role for ROS was further confirmed with the use of hydrogen peroxide (H_2_O_2_), an ROS-generating molecule [52]. The cells treated with H_2_O_2_ responded with an increased expression of CHOP, Zta, Ea-D, and cleaved caspase 3 (Figure 8A). An analysis of the kinetics of these events showed that CHOP expression occurred first, followed by lytic cycle proteins Zta and Ea-D, and cleaved caspase 3 (Figure 9). This sequence of events was strikingly similar to that seen in response to the treatment of cells with NEO212 (Figure 6B). In all of them, these mechanistic studies proposed a model where NEO212 generates ROS, which trigger ER stress and CHOP induction, resulting in the initiation of EBV’s lytic cycle. 

### 3.6. NEO212 Sensitizes C666.1 Cells to Ganciclovir

Having established NEO212 as a potent inducer of EBV’s lytic cycle, an obvious question arose as to whether these events would sensitize the tumor cells to the cytotoxic effects of GCV. Towards providing an answer, C666.1 cells were treated with increasing concentrations of NEO212 in the presence or absence of increasing concentrations of GCV, and long-term survival was determined by CFA. As presented in Figure 10, GCV at concentrations of 5, 10, or 20 µM had no inhibitory effect on these cells. However, when added in combination with NEO212, it potently enhanced the cytotoxic impact over treatment with NEO212 alone. For example, while 15 µM NEO212 reduced cell survival to 23.9%, the inclusion of 10 µM GCV reduced survival to 9.4% (Figure 10A). All cell survival percentages in response to the various drug combinations were analyzed for potential synergy/additivity/antagonism using the Bliss independence model (see details in Materials and Methods, Section 2.8). While combining the two drugs at their lowest concentrations (10 µM NEO212 plus 5 µM GCV) resulted in additive effects, the other 11 combinations showed a clear synergistic inhibitory impact on survival (Figure 10B). Together, these data show that GCV greatly enhances the cytotoxic impact of NEO212 and, conversely, that NEO212 enables GCV to unfold its inhibitory effects.

## 4. Discussion

The aim of our study was to determine whether NEO212 would be able to trigger the lytic cycle of EBV in NPC cells. The impetus for this project arose from an intriguing observation we made while studying the impact of NEO212 on EBV-positive strongly MGMT-positive C666.1 cells, where the IC50 of NEO212 was not influenced by the inclusion of the MGMT inhibitor O6BG. This was surprising for the following reasons. We had established that MGMT was strongly expressed in each NPC cell line we used (Figure 3). Furthermore, in the case of treatment with TMZ (Figure 1B), the inclusion of O6BG exquisitely chemosensitized the cells by at least an order of magnitude, confirming that MGMT was indeed functional and able to exert its expected protective task against the methylation of O6-guanine in all three cell lines. When treating EBV-negative cells (TW1 and TW4) with NEO212, the inclusion of O6BG was able to lower the IC50 (Figure 1A), although to a smaller degree than what had been observed with TMZ + O6BG. This differential has been observed with many other tumor cell types in the past: although NEO212 is always more potent than TMZ in MGMT-positive cells, the inclusion of O6BG nonetheless has consistently been found to moderately enhance its cytotoxic impact further, confirming that the methylation of O6-guanine plays a role in NEO212’s anticancer impact [39,40,43]. It was, therefore, quite unusual to see a complete lack of sensitization by O6BG in the NEO212-treated MGMT-positive C666.6 cells, where the IC50 of NEO212 was as low as it would be expected only in cells that are MGMT-negative (Figure 1A). This conjecture led us to investigate the hypothesis that, perhaps, the presence of latent EBV could play a role in this phenomenon.

During the latent stage of EBV, EBNA-1 is a constitutively expressed protein that may serve as an indicator of infection. We used it to confirm the presence of EBV in C666.1 cells and also in primary cells derived from a metastatic brain lesion of a patient with NPC (Figure 3). Zta is an immediate-early protein and Ea-D an early protein that appear during the lytic cycle but are not present during the latent stage [8,9]. They, therefore, can be used as reliable markers of entry into and progression through EBV’s lytic cascade. Our studies show that treatment with NEO212 resulted in the very pronounced appearance of both lytic markers, establishing NEO212 as a potent lytic cycle trigger (Figure 4). These effects were not restricted to in vitro conditions but, gratifyingly, could also be validated in NPC tumor tissue in vivo after the oral dosing of mice with NEO212 (Figure 6). Furthermore, both in vitro and in vivo, the emergence of Zta and Ea-D was followed by the appearance of cleaved (i.e., activated) caspase 3 (Figure 4 and Figure 6), demonstrating that the lytic events proceeded towards cell death.

In combination with the above-discussed issue regarding the ostensibly obsolete protective function of MGMT in NEO212-treated C666.1 cells, our finding that NEO212 potently triggered EBV’s lytic cycle suggested that the drug’s cytotoxic impact on these cells might be less derived from its ability to alkylate DNA but rather from its effect on the virus. C666.1 cells have been used extensively as a tool to study EBV and the lytic cycle; they are rare among the established NPC cell lines because they harbor native EBV genomes rather than EBV that was introduced during laboratory culture [46]. They do, however, entail the potential limitation that EBV’s lytic cycle is abortive and does not go towards terminal completion with virus release and cell lysis [9,53], which presumably was advantageous at the time when these cells were established as a continuous line; EBV-positive NPC cells from patients are notoriously difficult to establish in culture for the potential reason that this perturbing process might trigger cell lysis [54]. In any case, because of this recognized defect in C666.1 cells, it is reasonable to exclude virus-induced cell lysis as the mechanism of NEO212-induced cell killing. More likely, in view of the pro-survival and pro-tumorigenic functions of latent EBV proteins such as EBNA-1 and various viral latent membrane proteins (LMPs) [55,56,57], it is conceivable that these virus-derived stimuli become blunted during the lytic cycle, thereby sensitizing the cells towards apoptotic pathways [58]. In our in vivo model, treatment with NEO212 resulted in the pronounced down-regulation of EBNA-1 protein levels (Figure 6A). This observation is in support of the above-proposed model, as earlier reports have shown that EBNA-1 is critical for the expression of EBV’s transforming genes and for the survival of infected cells [59,60].

Among other interesting observations was that NEO212 potently stimulated the expression of CHOP both in vitro and in vivo, and this effect was detectable before the emergence of the earliest lytic marker, Zta (Figure 6). CHOP is a strongly pro-apoptotic protein that is induced by endoplasmic reticulum (ER) stress. Intriguingly, ER stress has been recognized within other contexts as a potential trigger of EBV’s lytic cycle. For example, the treatment of EBV-infected lymphoblastoid cells in vitro with classic ER stress inducers, such as thapsigargin and tunicamycin, was shown to stimulate the lytic cycle [61]. Bortezomib, a proteasome inhibitor and causative agent of ER stress, was shown to trigger the lytic cycle of EBV in Raji lymphoma cells and in B95-8 cells transformed in vitro by the virus [62]. In NPC cells, however, the effects of single-agent bortezomib are less clear, which, presumably, is reflective of the known cell type-specific differences in the activities of the various lytic cycle-inducing agents [5,10]. Similarly, clofoctol, an antibacterial antibiotic which stimulates ER stress, was shown to strongly trigger the lytic cycle in Raji lymphoma cells but substantially less efficiently in C666.1 cells [63]. A study by Hoji et al. [64] investigated the potency of several widely used ER stress inducers (thapsigargin, tunicamycin, brefeldin A, bortezomib, cyclosporine, dithiothreitol) in a panel of EBV-immortalized primary lymphoblastoid cell lines in vitro. Although all agents were able to stimulate ER stress (as determined by CHOP induction), only thapsigargin was also able to stimulate the lytic cycle (as determined by increased Zta expression and virus production). The authors concluded that, in their cell system, ER stress did not suffice to trigger the lytic cycle but that additional stimuli were required.

In our NPC model with C666.1 cells, NEO212 caused the robust induction of CHOP (Figure 6) along with the initiation of the lytic cycle. This correlation was further investigated by mechanistic studies where the knockdown of CHOP revealed that, without CHOP, NEO212 was unable to stimulate the lytic cycle (Figure 7). This requirement for CHOP was interesting for a variety of reasons. CHOP (also called C/EBPζ) is a transcription factor and a member of the CCAAT-enhancer-binding protein (C/EBP) family. Its six members are able to form homo- and hetero-dimers and bind to the CCAAT box motif present in a number of promoters, including within the Zta promoter of EBV [65,66]. It was, therefore, straightforward to postulate that CHOP would stimulate Zta transcription by binding to its promoter. While we have not yet investigated this hypothesis experimentally, it appears inconsistent with a report by Huang et al. [67], who studied the contribution of individual C/EBP proteins to lytic gene expression in EBV-infected AGS/BX1 NPC cells. The authors provided evidence that the stimulation of the lytic cycle with phorbol ester or sodium butyrate resulted in increased C/EBPα and C/EBPβ expression and that both proteins bound to multiple sites in the Zta promoter. However, the introduction of exogenous CHOP (C/EBPζ) into these cells antagonized the DNA-binding activity of the other C/EBP family members and reduced the Zta mRNA levels [67]. A study using the EBV-associated Burkitt’s lymphoma cell line Akata showed that C/EBPβ was also involved in lytic cycle induction by bortezomib [68]. However, as demonstrated by Hoji et al. [64] in EBV-immortalized primary lymphoblastoid cell lines, the potent induction of C/EBPβ expression by the ER stressor tunicamycin does not suffice to induce the lytic cycle. While our findings on the essential role of CHOP are, at least in part, in contrast to some of the above reports, we note a study by Zhou et al. [69] with murine gammaherpesvirus 68 (MHV68), providing evidence that ER stress-induced CHOP is essential in promoting B cell receptor-mediated lytic induction and virus production in murine SL-1 B cells. The sum of these studies further emphasizes the disparate responses of EBV within different cellular contexts and cautions against generalizations of the role of C/EBP transcription factors.

Previous studies investigating the role of ROS in triggering EBV’s lytic cycle yielded mixed and sometimes contradictory results [70]. For example, the treatment of Raji cells with H_2_O_2_ or FeSO_4_ was shown to stimulate the expression of Zta mRNA [71]. The same cells treated with chlorpyrifos, an organophosphate pesticide and known inducer of ROS, also enhanced Zta expression, and this effect could be ameliorated by incubation with the anti-oxidant N-acetylcysteine (NAC) [72]. The treatment of C666.1 cells with N-methyl-N’-nitro-N-nitrosoguanidine (MNNG) or H_2_O_2_ stimulated Zta and Ea-D expression, which could be prevented by the inclusion of ROS scavengers NAC, catalase, and reduced glutathione [73]. While these examples indicated a stimulatory role of ROS in EBV reactivation, other studies showed that these results cannot be generalized. For instance, experiments with Akata lymphoma cells showed that the treatment of cells with H_2_O_2_ or the H_2_O_2_-generating enzyme glucose oxidase effectively prevented the stimulation of the lytic cycle by B cell receptor signaling, prompting the authors to conclude that H_2_O_2_ was instrumental in the maintenance of EBV latency [74]. Using a panel of NPC cell lines, including C666.1, Hui et al. [75] demonstrated synergistic ROS induction by co-treatment with bortezomib and suberoylanilide hydroxamic acid (SAHA), a histone deacetylase inhibitor. However, despite pushing the cells into apoptosis, the increased ROS levels did not result in EBV reactivation. Moreover, although SAHA alone was able to trigger the lytic cycle in HA cells, co-treatment with bortezomib antagonized this stimulus. Interestingly, neither SAHA nor bortezomib caused EBV reactivation in C666.1 cells, once again providing an example of the pronounced differential impact of lytic cycle inducers on different cell types [75].

Our experiments investigating the contribution of ROS clearly indicated a positive role for these radicals. The treatment of C666.1 cells with H_2_O_2_ stimulated Zta and Ea-D expression (Figure 9), and the pre- or co-treatment of cells with anti-oxidants AA or BME effectively prevented NEO212-induced lytic cycle protein expression (Figure 8). Considering the above-discussed critical role of CHOP, we discovered that both anti-oxidants also blocked NEO212-induced CHOP expression as well as the emergence of activated caspase 3. These results are consistent with the known role of CHOP as a protein that is exquisitely inducible by a variety of cellular stressors, including oxidative stress [76,77]. Based on the entirety of our experimental results, we propose a model whereby the treatment of NPC cells with NEO212 causes increased ROS levels, which cause ER stress and induction of CHOP, resulting in the activation of EBV’s lytic cycle.

The ability of NEO212 to trigger the lytic cycle in EBV-infected NPC opens a new and unique treatment model for EBV-positive NPC. The current therapy for NPC consists of surgery (usually biopsy), radiation, and platinum chemotherapy. Each approach has well-known complications, including severe dysphagia from radiation and platinum-derived chemotherapeutic toxicity and resistance [78,79,80]. The ability of NEO212 to unleash the lytic cycle via ER stress not only allows the EBV to kill the host NPC cell and be released into the systemic circulation but also enables antiviral agents such as ganciclovir to kill it, leading to a complete destruction of both tumor and virus. Future work will be directed towards the temporal administration of NEO212 and ganciclovir using in vitro and in vivo EBV-infected NPC models. Other EBV-induced tumors such as Hodgkin’s lymphoma, non-Hodgkin’s lymphoma, gastric carcinoma, and post-transplant lymphoproliferative disorder (PTLD) are also potential candidates for this type of treatment.

Moreover, other virally induced tumors such as hepatitis B-induced hepatic carcinoma, human herpes virus 8 (HHV-8)-induced Kaposi’s sarcoma, and human papillomavirus (HPV)-induced cervical carcinoma [81,82] are all potential treatment models for lytic-induced tumoricidal therapy by NEO212. In each instance, infection of the host cell not only serves to induce tumorigenesis but also to shield the virus from antiviral therapy. As illustrated by bortezomib and its differential effects on EBV-infected cancer cells, we do not know what the direct effect of NEO212 will be among these other cancers until testing has been performed. However, the potential benefit of this type of treatment is definitely in the realm of possibilities and should be investigated.

## 5. Conclusions

This is the first study to demonstrate the ability of NEO212 to trigger the lytic cycle in EBV-positive NPC and sensitize these tumor cells to the antiviral drug GCV. Mechanistically, the NEO212 effects were mediated via ROS-stimulated CHOP induction, which proved essential for lytic cycle induction. We conclude that NEO212 holds promise for the improved treatment of EBV-positive NPC, in particular in the relapsed and refractory setting.

## Figures and Tables

**Figure 1 cancers-16-00936-f001:**
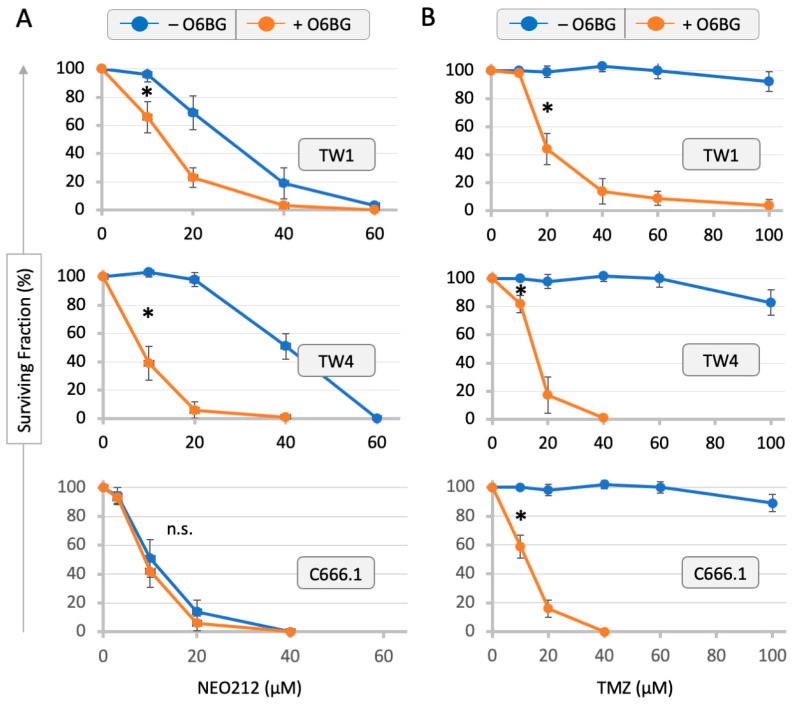
Inhibition of colony formation by NEO212 and TMZ can be enhanced by O6BG. Cells were treated with NEO212 or TMZ in the presence or absence of 15 µM O6BG, and the emerging colony formation was determined. (**A**) Cells treated with NEO212. (**B**) Cells treated with TMZ. Colony formation of untreated cells was set to 100%. Colony formation by vehicle-treated cells revealed no difference to untreated cells. Shown are the averages (n ≥ 3 ± SD). Asterisk (*) denotes statistically significant differences between treatment groups. n.s. = not significant.

**Figure 2 cancers-16-00936-f002:**
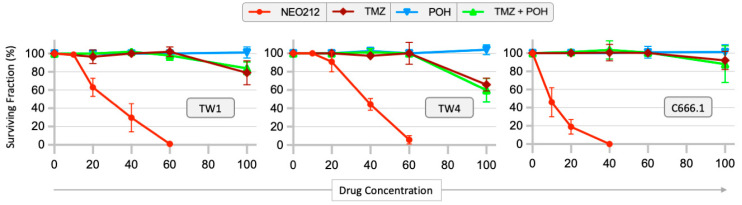
Inhibition of colony formation by NEO212 cannot be mimicked by TMZ + POH combination treatment. Cells were treated with NEO212, TMZ, POH, or TMZ mixed with POH at equimolar concentrations. The emerging colony formation was determined. The number of colonies from untreated cells was set to 100%. Colony formation by vehicle-treated cells revealed no difference to untreated cells. Shown are the averages (n ≥ 3 ± SD).

**Figure 3 cancers-16-00936-f003:**
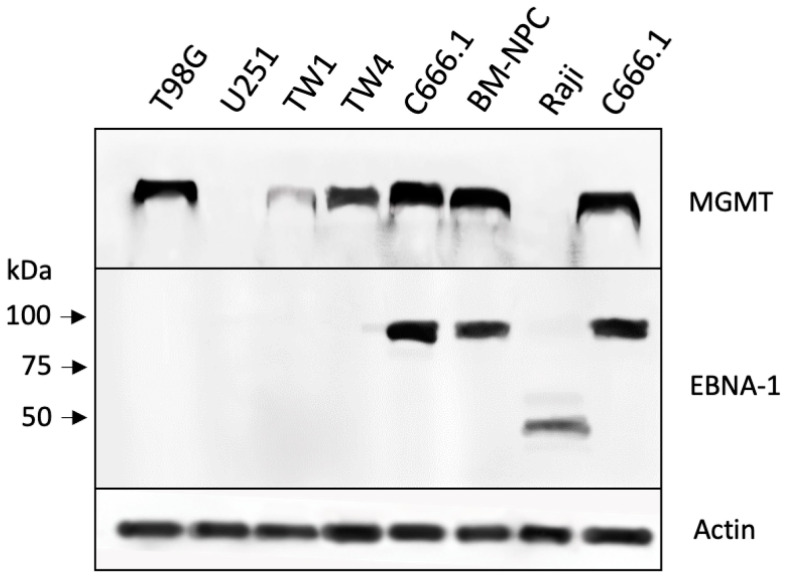
Differential expression levels of MGMT and EBNA-1 proteins. Protein lysates from several cultured cell lines were analyzed by Western blot for their basal level expression of MGMT and EBNA-1 protein. Actin was used as a loading control. T98G and U251 are glioblastoma cell lines that were included as positive and negative controls for MGMT expression, respectively. Raji is a Burkitt’s lymphoma cell line known to be positive for EBNA-1 (although known to express a variant of EBNA-1 protein with a different molecular weight [46]). Also included are primary NPC cells (BM-NPC) derived from the culture of the brain metastasis of a patient with advanced-stage NPC. Note that two different protein lysates derived from C666.1 cells were used: one from a low-confluency cell culture, the other from a high-density culture, showing that cell density did not impact MGMT or EBNA-1 expression levels.

**Figure 4 cancers-16-00936-f004:**
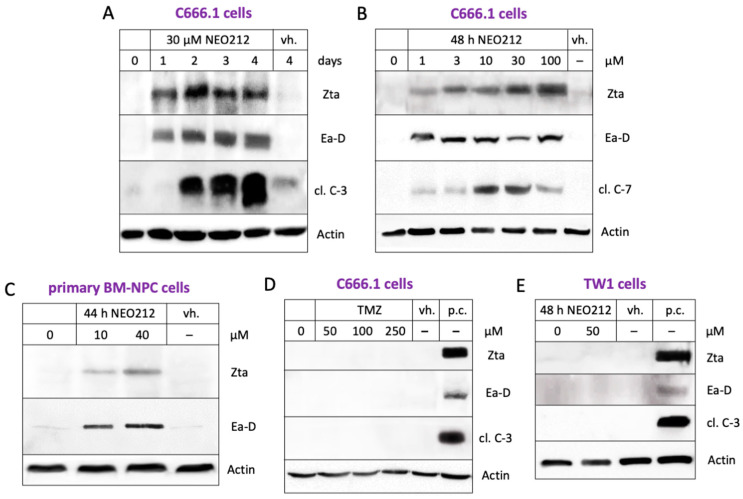
Stimulation of the lytic cycle by NEO212. Total protein lysates from cultured cells were analyzed by Western blot for markers of EBV lytic cycle (Zta; Ea-D) and for markers of apoptosis (cleaved fragments of caspases 3 and 7; cl. C-3; cl. C-7). Actin was used as a loading control. (**A**) C666.1 cells were treated for different lengths of time with NEO212. (**B**) C666.1 cells were treated with increasing concentrations of NEO212. (**C**) Primary BM-NPC cells (derived from the brain metastasis of a patient with NPC) were treated with NEO212. (**D**) C666.1 cells were treated with TMZ. (**E**) TW1 cells were treated with NEO212. vh.: lysate from vehicle-treated cells; p.c.: positive control.

**Figure 5 cancers-16-00936-f005:**
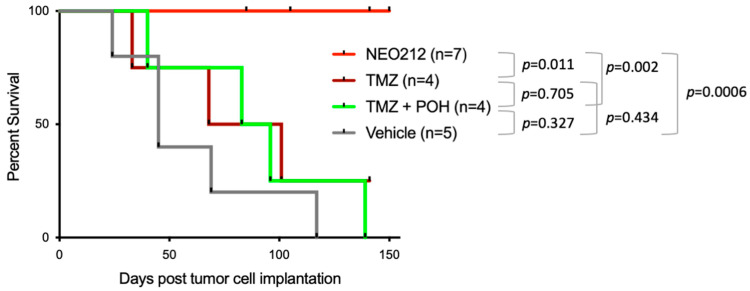
NEO212 prolongs survival of mice with xenografted C666.1 cells. Shown is a Kaplan–Meier survival plot. Mice with subcutaneously implanted C666.1 cells received treatment with 25 mg/kg NEO212, 25 mg/kg TMZ, 25 mg/kg TMZ mixed with 20 mg/kg POH, or a vehicle only. Treatment was administered via oral gavage once daily over the course of 5 days and then discontinued. Each animal was euthanized once the tumor size reached the maximum allowable size of 1500 mm^3^. Within the group of NEO212-treated mice, three animals had to be euthanized due to injuries from bite wounds that were unrelated to tumor growth; therefore, these mice were censored (indicated by black tick marks on days 85, 105, and 141). Similarly, the last remaining mouse in the group of TMZ-treated animals had to be censored on day 141; black tick mark). Statistical comparison between groups was performed with a log-rank test; the *p*-values are shown.

**Figure 6 cancers-16-00936-f006:**
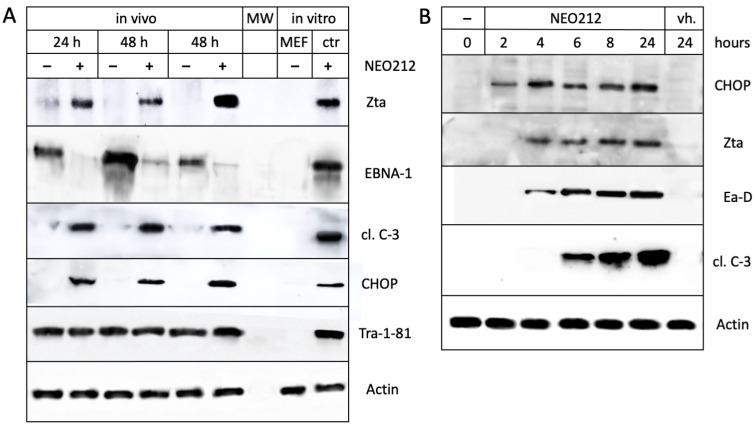
Stimulation of the lytic cycle by NEO212 in vivo and in vitro, along with ER stress. (**A**) Mice with subcutaneous C666.1 tumors were treated with NEO212 or a vehicle. Twenty-four hours later, one pair of mice was euthanized; forty-eight hours later two pairs of mice were euthanized. Tumor tissue was collected, and the derived total protein lysates were analyzed by Western blot for the presence of EBNA-1 and lytic marker Zta, along with apoptosis marker cleaved caspase 3 (cl. C-3) and ER stress marker CHOP. Tra-1-81 is a human-specific epitope not present in mouse cells; this was confirmed by including a lysate from mouse embryo fibroblasts (MEF). Actin was used as a loading control. ctr.: positive control (cultured C666.1 cells treated with NEO212). (**B**) Cultured C666.1 cells were treated with 25 µM NEO212 and harvested at different time points thereafter. Protein lysates were analyzed by Western blot. Note that the CHOP protein levels increase before the emergence of lytic markers Zta and Ea-D.

**Figure 7 cancers-16-00936-f007:**
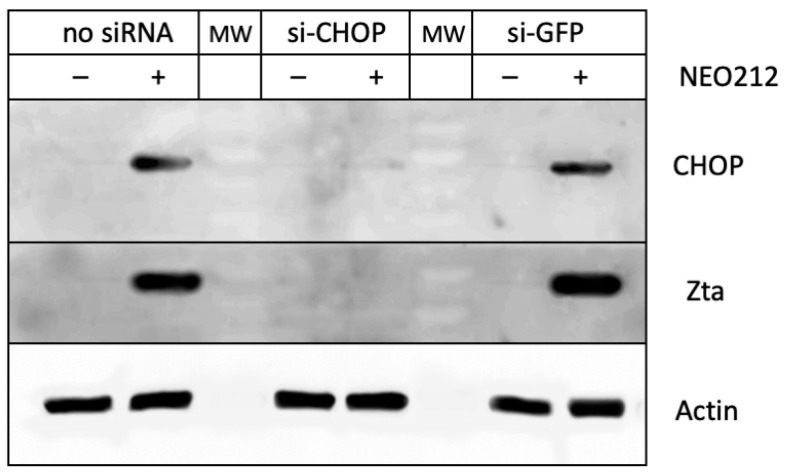
Knockdown of CHOP prevents the induction of the lytic cycle. C666.1 cells were transfected with siRNA targeting CHOP. Control cells received siRNA against green fluorescent protein (GFP) or remained untransfected. Each culture was distributed into two separate culture dishes and treated with 30 µM NEO212 or a vehicle for 24 h, followed by Western blot analysis. Lanes labeled MW contain a molecular weight marker.

**Figure 8 cancers-16-00936-f008:**
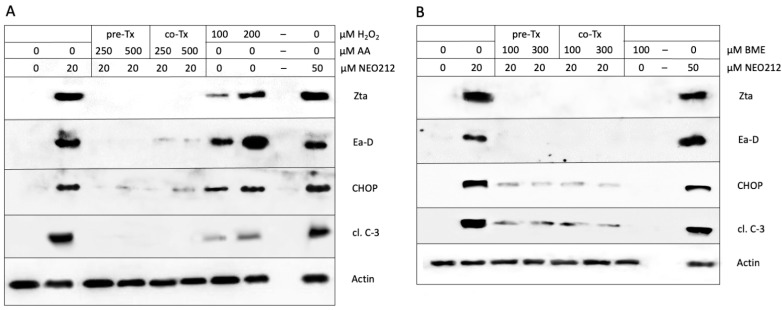
ROS mediate NEO212-induced ER stress and the lytic cycle. (**A**) Cells were treated with NEO212 in the presence or absence of ascorbic acid (AA). Two modes of AA treatment were used: (i) AA was administered to cells overnight, and the next day NEO212 was added together with a fresh dose of AA (pre-Tx); (ii) NEO212 was added together with AA, without AA pre-treatment (co-Tx). Separately, cells were also exposed to hydrogen peroxide (H_2_O_2_). Twenty-four hours later, cells were harvested and analyzed by Western blot. (**B**) Cells were treated with NEO212 in the presence or absence of beta-mercaptoethanol (BME), similar to the schedule outlined for AA above.

**Figure 9 cancers-16-00936-f009:**
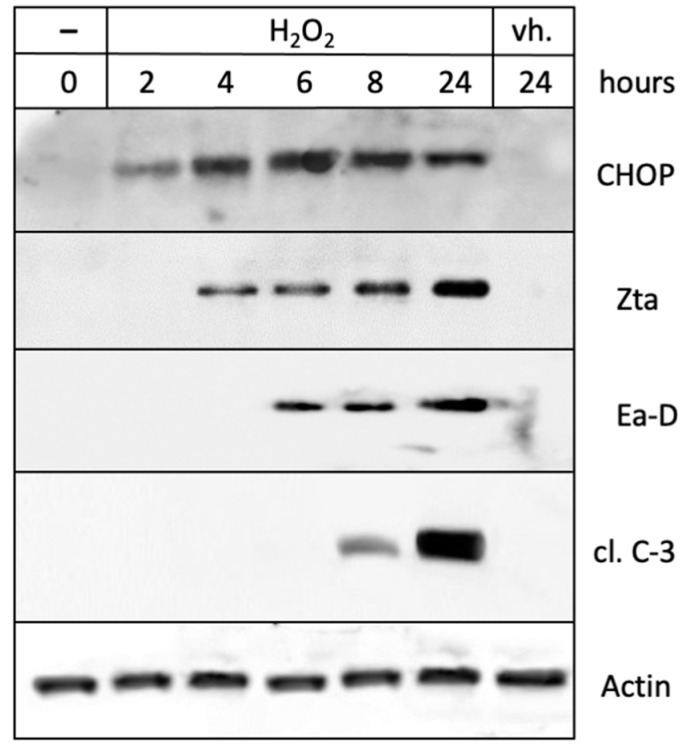
Hydrogen peroxide triggers ER stress and the lytic cycle. C666.1 cells were treated with 200 µM H_2_O_2_ for different lengths of time. Total cell lysates were analyzed by Western blot.

**Figure 10 cancers-16-00936-f010:**
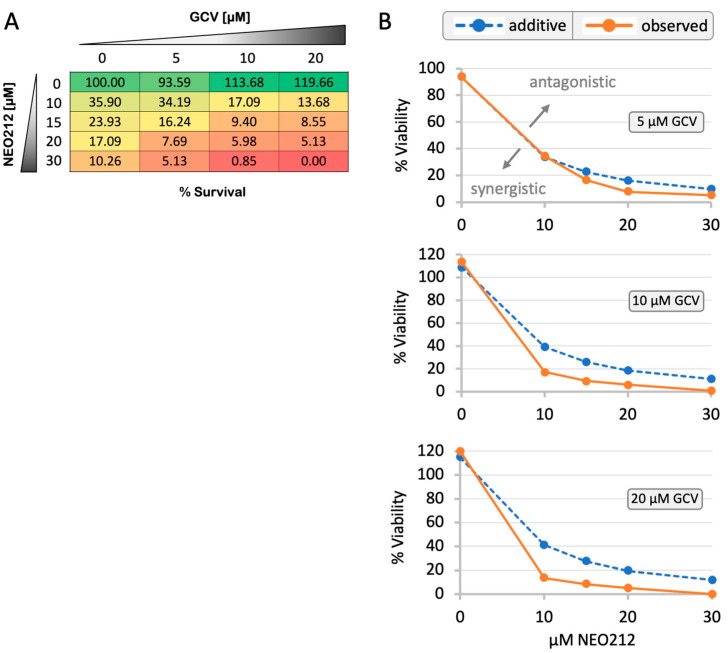
NEO212 synergizes with ganciclovir. C666.1 cells were treated with NEO212 alone, ganciclovir (GCV) alone, or both in combination. (**A**) Numbers indicate percent survival under each treatment condition, as determined by colony formation assays. Note that GCV alone did not reduce survival but greatly enhanced the cytotoxic impact of NEO212. (**B**) Statistical analysis of drug combination effects of NEO212 plus GCV. The dotted blue line represents the predicted (i.e., calculated) combination effect (Y_P_) based on the Bliss independence model [48], which signifies an additive effect. The orange straight line shows the actually observed (i.e., measured) combination response (Y_O_). Overlap of the blue data point with the orange data point in the top graph (10 µM NEO212 combined with 5 µM GCV) indicates an additive effect. However, all other data points in the three graphs result in the separation of the lines and a pronounced left-shift of the orange line, indicating strong synergistic effects.

## Data Availability

All data are contained within this article.

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
