# Peer review of "Activation of Epstein–Barr Virus’ Lytic Cycle in Nasopharyngeal Carcinoma Cells by NEO212, a Conjugate of Perillyl Alcohol and Temozolomide"

_cancers, 2024, doi:10.3390/cancers16050936_

Round 1
Reviewer 1 Report
Comments and Suggestions for Authors
General Comments
The manuscript by Hannah Hartman-Houstman and colleagues presents a compelling study on the efficacy of NEO212 in inducing the lytic cycle of Epstein-Barr Virus (EBV) in Nasopharyngeal Carcinoma (NPC) cells. The use of both in vitro and in vivo models, including primary patient-derived cells, is commendable. The findings that NEO212 prompts the production of immediate-early and early-stage lytic viral proteins, and the involvement of CHOP and Reactive Oxygen Species (ROS), are significant contributions to the field. This study adds valuable insights into potential therapeutic strategies for NPC.
Specific Comments
1. Figure 3 Clarification: The presence of two sets of C666.1 data in Figure 3 raises a question. Could you please clarify if this repetition is intentional for comparative purposes or an oversight? If it's intentional, a brief explanation in the figure legend or text would enhance the understanding of the figure.
2. Molecular Weight Labeling in Western Blots: For improved clarity, I recommend labeling the molecular weights in each Western blot image. This addition would assist in the accurate interpretation of the protein bands and support the conclusions drawn from these results.
Conclusion
The manuscript is well-structured and presents its findings clearly. With the suggested minor revisions, particularly in terms of data presentation, I believe it will be a valuable contribution to the literature. The study is poised for acceptance after these revisions.
Author Response
The authors appreciate the time and effort by this reviewer to critically read the manuscript and provide thoughtful comments for improvement. Thank you.
Reviewer 1
General Comments
The manuscript by Hannah Hartman-Houstman and colleagues presents a compelling study on the efficacy of NEO212 in inducing the lytic cycle of Epstein-Barr Virus (EBV) in Nasopharyngeal Carcinoma (NPC) cells. The use of both in vitro and in vivo models, including primary patient-derived cells, is commendable. The findings that NEO212 prompts the production of immediate-early and early-stage lytic viral proteins, and the involvement of CHOP and Reactive Oxygen Species (ROS), are significant contributions to the field. This study adds valuable insights into potential therapeutic strategies for NPC.
Specific Comments
- Figure 3 Clarification: The presence of two sets of C666.1 data in Figure 3 raises a question. Could you please clarify if this repetition is intentional for comparative purposes or an oversight? If it's intentional, a brief explanation in the figure legend or text would enhance the understanding of the figure.
Response: Yes, there are two lanes with lysates from C666.1 cells included in this Western blot. This was for validation purposes, as the C666.1 cells are the most critical for the current study. One set of cells was harvested a low cell confluency, the other at high cell density. We wanted to ascertain that cell density had no impact on the expression of the proteins we were analyzing. We have now added this explanation to the figure legend.
- Molecular Weight Labeling in Western Blots: For improved clarity, I recommend labeling the molecular weights in each Western blot image. This addition would assist in the accurate interpretation of the protein bands and support the conclusions drawn from these results.
Response: Yes, agreed. We had already provided this information in the Supplementary Material. Adding molecular weights to uncropped images allows for the most accurate evaluation of the signals of a given blot. We therefore provided all uncropped Western blot images in the Supplementary Material, along with molecular weight distribution, as well as data from densitometric scanning (which also is useful for interpretation of results).
Conclusion
The manuscript is well-structured and presents its findings clearly. With the suggested minor revisions, particularly in terms of data presentation, I believe it will be a valuable contribution to the literature. The study is poised for acceptance after these revisions.
Response: We appreciate the constructive feedback by the reviewer.
Reviewer 2 Report
Comments and Suggestions for Authors
Here are some comments and suggestions for the authors:
Clarity in Introduction: Enhance the introduction by providing more detailed background, particularly on the mechanisms of NEO212's interaction with the EBV lytic cycle. This could help in setting clearer expectations for the study's outcomes.
Methodology Details: Consider adding more detailed descriptions in the methodology section, especially regarding experimental conditions, to improve reproducibility.
Statistical Analysis: Expand the section on statistical analysis to include more details about the tests used and criteria for significance. This would strengthen the validity of your results.
Comparative Analysis: Include a comparative analysis of NEO212's efficacy against existing treatments or agents, which could provide context for its potential clinical utility.
Addressing Limitations: Acknowledge any limitations or potential biases in your study, such as the selection of specific cell lines or the extrapolation of in vitro findings to clinical scenarios.
Broader Implications: Elaborate on the broader implications of your findings for cancer therapy, especially for EBV-related cancers, and suggest future research directions.
Figures and Tables: Ensure all graphical data and tables are clear, accurately labeled, and effectively support the text.
Reference Check: Verify all references for their currency and relevance, ensuring the manuscript is grounded in the most recent research.
Ethical Considerations: If not already done, discuss any ethical considerations related to your study.
Language and Style Review: Consider a professional language review for clarity, grammar, and style, to align with high publication standards.
These suggestions aim to enhance the impact and readability of your manuscript, ensuring effective communication of its significant findings.
Comments on the Quality of English LanguageThe manuscript is well-written in English with a good standard of academic language appropriate for a scientific publication. The sentences are structured clearly, the terminology is specific and accurate, and the overall readability is high. This level of English is suitable for the intended audience of researchers and professionals in the field. However, like any scientific manuscript, it could potentially benefit from a professional language review to ensure clarity, grammar, and style align with publication standards. This step is often helpful for refining the manuscript's language to the highest professional level.
Author Response
The authors appreciate the time and effort by this reviewer to critically read the manuscript and provide thoughtful comments for improvement. Thank you.
Reviewer 2
Here are some comments and suggestions for the authors:
Clarity in Introduction: Enhance the introduction by providing more detailed background, particularly on the mechanisms of NEO212's interaction with the EBV lytic cycle. This could help in setting clearer expectations for the study's outcomes.
Response: Our study is the first to describe an interaction of NEO212 with EBV’s lytic cycle. No other studies on this topic exist; therefore, we are unable to provide more detailed background on these mechanisms. The mechanisms are presented, for the very first time, in the Results section, and their implications are discussed in the Discussion section. Nonetheless, we have now included additional details and background in the Introduction, primarily pertaining to the mechanism by which triggering the lytic cycle can sensitize tumor cells to ganciclovir.
Methodology Details: consider adding more detailed descriptions in the methodology section, especially regarding experimental conditions, to improve reproducibility.
Response: It is not clear to us which parts of the methodology section would require further details in this reviewer’s opinion. Nonetheless, the methodology section has now been expanded.
Statistical Analysis: Expand the section on statistical analysis to include more details about the tests used and criteria for significance. This would strengthen the validity of your results.
Response: Now expanded.
Comparative Analysis: Include a comparative analysis of NEO212's efficacy against existing treatments or agents, which could provide context for its potential clinical utility.
Response: In the Discussion section, we do have an extensive description and critical analysis of other agents known to trigger the lytic cycle of EBV, as compared to NEO212.
Addressing Limitations: Acknowledge any limitations or potential biases in your study, such as the selection of specific cell lines or the extrapolation of in vitro findings to clinical scenarios.
Response: In the Discussion, we have addressed the limitations of the C666.1 cell line that we used (although please note that primary patient-derived, EBV-positive cells were included in our study as well and generated similar results). A clinical outlook is provided as well, and clearly labeled as “potential”: meaning that studies need to be done before conclusions can be drawn.
Broader Implications: Elaborate on the broader implications of your findings for cancer therapy, especially for EBV-related cancers, and suggest future research directions.
Response: Thank you. Please see the last two paragraphs of the Discussion section.
Figures and Tables: Ensure all graphical data and tables are clear, accurately labeled, and effectively support the text.
Response: Done. Thank you.
Reference Check: Verify all references for their currency and relevance, ensuring the manuscript is grounded in the most recent research.
Response: Done. Thank you.
Ethical Considerations: If not already done, discuss any ethical considerations related to your study.
Response: Ethical statements and reference to IACUC and IRB reviews are included.
Language and Style Review: Consider a professional language review for clarity, grammar, and style, to align with high publication standards.
Response: Done. Thank you.
These suggestions aim to enhance the impact and readability of your manuscript, ensuring effective communication of its significant findings.
Response: Great suggestions! We appreciate the constructive review and feedback from this reviewer.
Comments on the Quality of English Language
The manuscript is well-written in English with a good standard of academic language appropriate for a scientific publication. The sentences are structured clearly, the terminology is specific and accurate, and the overall readability is high. This level of English is suitable for the intended audience of researchers and professionals in the field. However, like any scientific manuscript, it could potentially benefit from a professional language review to ensure clarity, grammar, and style align with publication standards. This step is often helpful for refining the manuscript's language to the highest professional level.
Response: Thank you again, your suggestions were heeded.
Reviewer 3 Report
Comments and Suggestions for Authors
Hartman-Houstman et al., in their paper describe a novel compound, NEO212, which is a bio-conjugate of two existing anti-cancer drugs, that can be used to treat to treat nasopharyngeal carcinoma (NPC) cells that harbor Epstein-Barr virus (EBV), commonly associated with NPC. It is known that EBV shelters in cancer cells where it remains in the latent state. In this latent state, the virus thrives and at the same time allows the cancer cell to grow. The two compounds from which NEO212 is derived, namely TMZ and POH have shown some limitations as anticancer drugs. In this manuscript provided here, the authors have introduced NEO212 as a potential compound that can target EBV positive NPC cells that can induce EBV to undergo lysis and in the process kill the cancer cells that harbor the virus.
This is a very well-designed, systematic study and the results in the manuscript are well presented. The authors have done an in-depth study to provide a mechanism of NEO212 activity where they show that NEO212 generates reactive oxygen species generation that activates CHOP and the overall ER stress response mediated by the cell leads to EBV lytic cycle activation and death or reduction in tumor cell mass both in vitro and in vivo. One concern that has been addressed in the discussion is the release of EBV in the bloodstream upon its lytic cycle activation. The authors mention the use of an antiviral agent such as ganciclovir to treat the virus. It would be of great value if the authors could provide data on this in their in vivo model.
Author Response
The authors appreciate the time and effort by this reviewer to critically read the manuscript and provide thoughtful comments for improvement. Thank you.
Reviewer 3
Comments and Suggestions for Authors
Hartman-Houstman et al., in their paper describe a novel compound, NEO212, which is a bio-conjugate of two existing anti-cancer drugs, that can be used to treat to treat nasopharyngeal carcinoma (NPC) cells that harbor Epstein-Barr virus (EBV), commonly associated with NPC. It is known that EBV shelters in cancer cells where it remains in the latent state. In this latent state, the virus thrives and at the same time allows the cancer cell to grow. The two compounds from which NEO212 is derived, namely TMZ and POH have shown some limitations as anticancer drugs. In this manuscript provided here, the authors have introduced NEO212 as a potential compound that can target EBV positive NPC cells that can induce EBV to undergo lysis and in the process kill the cancer cells that harbor the virus.
This is a very well-designed, systematic study and the results in the manuscript are well presented. The authors have done an in-depth study to provide a mechanism of NEO212 activity where they show that NEO212 generates reactive oxygen species generation that activates CHOP and the overall ER stress response mediated by the cell leads to EBV lytic cycle activation and death or reduction in tumor cell mass both in vitro and in vivo. One concern that has been addressed in the discussion is the release of EBV in the bloodstream upon its lytic cycle activation. The authors mention the use of an antiviral agent such as ganciclovir to treat the virus. It would be of great value if the authors could provide data on this in their in vivo model.
Response: The suggestion regarding ganciclovir is constructive and appreciated. We have now added new data with the use of ganciclovir to the manuscript (see newly added Figure 10), along with corresponding text in the Results section and throughout. Thank you for reading our manuscript and the thoughtful comments.
Reviewer 4 Report
Comments and Suggestions for Authors
The Epstein-Barr Virus (EBV) stands as the first and one of the most extensively studied human tumor viruses. Infection with EBV has been identified as either causal or associated with numerous human malignancies. Conventional anti-cancer strategies have shown limited effectiveness, prompting exploration into viral oncolytic therapy as an alternative. However, a key challenge lies in the scarcity of potent viral lytic-inducing agents that exhibit minimal toxicity. The manuscript by Hartman-Houstman et al. details their efforts to investigate novel triggers for EBV lytic activation.
Comments:
1. Does NEO212, as a conjugate between TMZ and POH, still retain the ability to methylate O6-guanine?
2. The title of the Y-axis in Figures 1 and 2 is confusing; consider replacing "surviving rate of colony" with "relative fold of colony number" or a similar term.
3. In Figure 3, what was the purpose of the duplicated loading of C666.1 whole-cell lysate for Western blot?
4. For Figure 4A and 4B, why were caspase 3 and 7 examined separately? Additionally, could the authors elaborate on why the early lytic protein EA-D levels did not exhibit a similar increasing trend as ZTA when cells were treated with increased doses of NEO212?
5. The format of Figure 5 is confusing. The mice did not succumb to tumor growth but rather declined when the tumor size reached 1,500 mm3. Consider adjusting the representation to accurately reflect the data in a survival curve.
6. In Figure 6A, a concern arises from the Western blot showing the actin antibody recognizing both human and mouse actin. This complicates the quantification of protein levels in C666.1 based on actin normalization. Please add the molecular weight of the protein ladder to each Western blot figure.
7. The potent knockdown of CHOP in C666.1 is notable. It would be interesting to assess transfection efficiency using a fluorescein-conjugated scramble siRNA.
Author Response
The authors appreciate the time and effort by this reviewer to critically read the manuscript and provide thoughtful comments for improvement. Thank you.
Reviewer 4
The Epstein-Barr Virus (EBV) stands as the first and one of the most extensively studied human tumor viruses. Infection with EBV has been identified as either causal or associated with numerous human malignancies. Conventional anti-cancer strategies have shown limited effectiveness, prompting exploration into viral oncolytic therapy as an alternative. However, a key challenge lies in the scarcity of potent viral lytic-inducing agents that exhibit minimal toxicity. The manuscript by Hartman-Houstman et al. details their efforts to investigate novel triggers for EBV lytic activation.
Comments:
- Does NEO212, as a conjugate between TMZ and POH, still retain the ability to methylate O6-guanine?
Response: Yes, it does. This has been documented in several of our previously published studies. They are cited in the manuscript.
- The title of the Y-axis in Figures 1 and 2 is confusing; consider replacing "surviving rate of colony" with "relative fold of colony number" or a similar term.
Response: We are not quite sure what the reviewer is referring to, as these y-axes are not labeled “surviving rate of colony”. They are labeled “Surviving Fraction [%]”, which represents a commonly used term.
- In Figure 3, what was the purpose of the duplicated loading of C666.1 whole-cell lysate for Western blot?
Response: Yes, there are two lanes with lysates from C666.1 cells included in this Western blot. This was for validation purposes, as the C666.1 cells are the most critical for the current study. One set of cells was harvested a low cell confluency, the other at high cell density. We wanted to ascertain that cell density had no impact on the expression of the proteins we were analyzing. We have now added this explanation to the figure legend.
- For Figure 4A and 4B, why were caspase 3 and 7 examined separately? Additionally, could the authors elaborate on why the early lytic protein EA-D levels did not exhibit a similar increasing trend as ZTA when cells were treated with increased doses of NEO212?
Response: In several of our experiments, we analyzed different caspases as general markers of apoptosis. In most cases, drug-induced activation of different caspases (3, 4, 7) was comparable. While we mostly show cleaved C-3 as the prime example throughout the study, we wanted to add one result with cleaved C-7, just as an example that not only C-3 responds to drug treatment.
We do not know why EA-D does not show the same increasing trend as ZTA in Figure 4B. We can only speculate that the activity of transcription factor ZTA already is maximal at lower expression levels, i.e., that high ZTA levels are not required for full expression of EA-D. Also, EA-D expression is not only dependent on ZTA; other factors are involved. For example, it is known that ZTA and RTA can act synergistically to stimulate the expression of other EBV genes.
- The format of Figure 5 is confusing. The mice did not succumb to tumor growth but rather declined when the tumor size reached 1,500 mm3. Consider adjusting the representation to accurately reflect the data in a survival curve.
Response: We understand the reviewer’s comment, but do not see a better option to present these survival data. We deem our presentation as appropriate and don’t think there is a better way of graphing these results. As per IACUC regulations at our institution, a tumor size of 1,500 mm3 is considered a humane endpoint and mice must be euthanized. This is not unlike the situation with intracranially implanted tumor cells, for example, where mice must be euthanized based on a 5-point scale that measures behavior (neurological deficits), along with general appearance and health. We realize that in both cases, it is not the tumor that kills the animal, but rather extraneous signs that point to the detrimental impact of tumor growth that necessitates euthanasia.
- In Figure 6A, a concern arises from the Western blot showing the actin antibody recognizing both human and mouse actin. This complicates the quantification of protein levels in C666.1 based on actin normalization. Please add the molecular weight of the protein ladder to each Western blot figure.
Response: The molecular weight of the protein ladder for all Western blots is shown in the Supplemental Material. We had already provided this information in the Supplementary Material. Adding molecular weights to uncropped images allows for the most accurate evaluation of the signals of a given blot. We therefore provided all uncropped Western blot images in the Supplementary Material, along with molecular weight distribution, as well as data from densitometric scanning (which also is useful for interpretation of results).
Regarding the quantification and normalization of C666.1 protein levels: Please note that we used Tra-1-81 for this. Tra-1-81 specifically recognizes human cells (C666.1) and does not cross-react with mouse cells (MEF), as demonstrated in Figure 6A. Actin was merely used as a control for overall protein loading.
- The potent knockdown of CHOP in C666.1 is notable. It would be interesting to assess transfection efficiency using a fluorescein-conjugated scramble siRNA.
Response: Yes, transfection efficiency could be of interest. However, in our knockdown experiment, it is not needed for the readout and the interpretation of the results. Transfection efficiency was high enough to knock down CHOP below detectable levels.
Round 2
Reviewer 4 Report
Comments and Suggestions for Authors
The revised version of manuscript addressed most of reviewer's comments, although the evidence of completion of EBV lytic cascade has not been provided.